# Advancing towards cancer theragnostic by probing the $^{225}$Ac decay chain with ultra-high-resolution metallic magnetic calorimeter based detectors

Kiara Maurer [1,8], Daniel Unger [2,8], Martin Behe [3], Andreas Knecht[3], Katharina von Schoeler [4], Daniel Hengstler[2], Tonya Vitova [1], Martina Benešová-Schäfer [5], Andreas Fleischmann[2] ✉, Loredana Gastaldo[2], Carmen Wängler [6] ✉, Christian Enss [2,7] ✉ & Bianca Schacherl [1] ✉

## Abstract

**Background** $^{225}$Ac is one of the most promising short-lived radionuclides for application in targeted alpha therapy (TαT). However, quantitative detection of $^{225}$Ac and its daughter radionuclides by γ-spectroscopy has so far been unattainable. Achieving this would enable precise organ dosimetry and better targeting of the therapeutic potential of the $^{225}$Ac decay series.

**Methods** In this study, an X-ray and γ-spectrum of an $^{225}$Ac sample (89 kBq) was recorded with a Metallic Magnetic Calorimeter (MMC) detector over a wide energy range of 5 to 125 keV.

**Results** Here, we show the feasibility of detecting γ-lines of $^{225}$Ac, $^{221}$Fr, $^{213}$Bi, and $^{209}$Tl with ultra-high-resolution spectroscopy (FWHM of 23 eV @ 5.9 keV). By analyzing characteristic X-ray fluorescence lines, it is also possible to distinguish $^{221}$Fr, $^{217}$At, $^{213}$Bi, $^{213}$Po, and $^{209}$Pb in this sample. Additionally, particle induced X-ray emission from $^{225}$Ac is found.

**Conclusions** To the best of our knowledge, this first MMC based X-ray and γ-spectrum of $^{225}$Ac demonstrates the detector's potential, as it enables the separate detection of most nuclides in the $^{225}$Ac decay chain (all except $^{221}$Ra and $^{217}$Rn) with high sensitivity, excellent energy resolution, and precise energy calibration, paving the way for key technological advancements in research and clinical applications in nuclear medicine.

## Plain language summary

$^{225}$Ac is a radioactive material. It releases tiny high-energy particles that can destroy cancer cells very precisely, and because of that, it can be used for a cancer treatment called "targeted alpha therapy." During targeted alpha therapy, $^{225}$Ac can decay into various other products that can distribute throughout the body and cause side effects. It is important to be able to differentiate these decay products from the intended $^{225}$Ac, but current methods can only detect two of the many $^{225}$Ac decay products. In this study, we analyzed $^{225}$Ac using a special measurement device so called "Metallic Magnetic Calorimeters." Using this method allows us to identify $^{225}$Ac and more decay products than previously reported, and therefore further allows us to track the products of the main decay chain of $^{225}$Ac. Continued work in this area could support future research and improve medical uses of $^{225}$Ac.

Targeted alpha therapy (TαT) is a promising treatment method in which highly energetic alpha particles are selectively delivered to cancer cells, destroying them while minimizing damage to healthy surrounding tissues in the tumor environment[1]. The most promising candidates for therapeutic application in TαT currently include $^{225}$Ac, $^{223}$Ra, $^{227}$Th, or $^{213}$Bi[2]. The mass of an alpha particle is ~8000 times higher than the mass of a beta particle, and

alpha decay releases large amounts of energy over short distances of 50–100 μm. Due to the high linear energy transfer (LET ~ 100 keV/μm), a more specific destruction of tumor cells is possible[2].

$^{225}$Ac has a half-life of 9.9 days and decays via α-emission in more than 99% of cases. Its decay chain, which contains four successive alpha decays, is shown in Fig. 1. Although $^{225}$Ac is currently the most promising candidate

¹Karlsruhe Institute of Technology (KIT), Institute for Nuclear Waste Disposal (INE), Karlsruhe, Germany. ²Kirchhoff Institute for Physics, Heidelberg University, Heidelberg, Germany. ³Paul Scherrer Institute (PSI), Villigen, Switzerland. ⁴ETH Zürich, Institute for Particle Physics and Astrophysics, Zürich, Switzerland. ⁵German Cancer Research Center (DKFZ), Research Group Translational Radiotheranostics, Heidelberg, Germany. ⁶Clinic of Radiology and Nuclear Medicine, Biomedical Chemistry, Medical Faculty Mannheim of Heidelberg University, Mannheim, Germany. ⁷Karlsruhe Institute of Technology (KIT), Institute for Electronics and Data Processing (IPE), Karlsruhe, Germany. ⁸These authors contributed equally: Kiara Maurer, Daniel Unger. ✉e-mail: Andreas.Fleischmann@kip.uni-heidelberg.de; Carmen.Waengler@medma.uni-heidelberg.de; christian.enss@kip.uni-heidelberg.de; bianca.schacherl@kit.edu

**Fig. 1 | Decay chain of $^{225}$Ac and its daughter radionuclides, including four subsequent α-decays (4α2β−).** The daughter nuclides are: $^{221}$Fr, $^{221}$Ra, $^{217}$At, $^{217}$Rn, $^{213}$Bi, $^{213}$Po, $^{209}$Tl, $^{209}$Pb, and $^{209}$Bi (blue: β−-decay; yellow: α-decay).

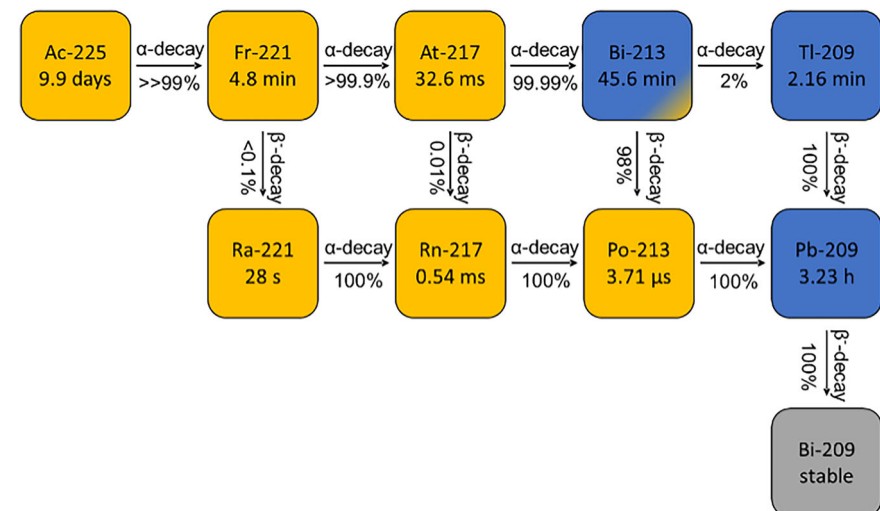

for TαT, there are still limitations. A chelator complexes the radionuclide in a radiopharmaceutical and thereby enables its targeted delivery. However, during decay, the daughter nuclides such as $^{221}$Fr have altered chemical bonding interaction, suboptimal chelation and hence liberation of the radiometal ion. Further, the energy released and the recoil effects during the alpha decay can break the chemical bonds. Thus, the liberated radionuclide can migrate to undesired locations in various organs, resulting in a significant dose burden to healthy organs and tissues. To understand the behavior of all free radionuclides in the body, both quantitative and qualitative detection of $^{225}$Ac and all its daughter radionuclides is desirable.

This information is essential for determining radionuclide distribution and doses to organs and tissues, which are the prerequisite for adequate treatment planning and the determination of the patient-specific optimal amount of activity for treatment. This is mandatory in order to maximize the dose applied to the tumor while limiting toxicity to organs that should be spared.

Understanding radionuclide distribution may help to design more efficient ways to deliver or retain these radionuclides. State-of-the-art methods for visualizing the distribution of radionuclides include Positron Emission Tomography (PET), which detects the gamma rays of the positron β+β−-annihilation of 511 keV, enabling detection of positron-emitting nuclides. Additionally, Single-Photon Emission Computed Tomography (SPECT) detects strong γ-lines in a high energy range and is therefore applicable to broader range of radionuclides. However, with the current SPECT detection methods, only $^{221}$Fr and $^{213}$Bi are imageable in the $^{225}$Ac decay chain since they emit γ-rays with suitably high energies and substantial emission probability (218 keV with $I$ = 11.4% and 440 keV with $I$ = 25.9%). Even this is challenging because the quantities used for TαT are small, making SPECT difficult to perform. Moreover, the detectors, mostly sodium iodide (NaI) and cesium iodide (CsI) thallium-doped scintillators[3], applied in SPECT suffer from low energy resolution. This results in a broad main peak at about 78 keV, in an $^{225}$Ac spectrum, which effectively consists of different γ-lines of $^{225}$Ac and the various daughter radionuclides ($^{221}$Fr and $^{213}$Bi)[4]. In addition to the low energy resolution, the spatial resolution is limited by the aperture size of the collimator to a few millimeters[3].

In comparison to NaI scintillation detectors[5] used in SPECT a High Purity Germanium (HPGe) detector has a higher energy resolution of about 0.5 keV @ 5.9 keV and 0.75 keV @ 122 keV[6–8]. A disadvantage is the limited efficiency (<10%)[9,10]. In the energy range below 100 keV[4], peaks overlap, resulting in peaks that merge into a large background. Additionally, pileup, where multiple γ-photon events occur too close together for the system to distinguish, lead to overlapping signals that increase the apparent background as well and distort the energy spectrum. This can result in spurious counts and reduced energy resolution[11], leading to an inability to differentiate γ-lines from different radionuclides like $^{213}$Bi, $^{221}$Fr and $^{225}$Ac in the low-energy region (<100 keV)[12,13].

An even better energy resolution of ~125 eV @ 5.9 keV $MnK_\alpha$ line is achieved by Silicon Drift Detectors (SDDs)[14]. The incident radiation leads to ionization processes; the resulting high-energy photoelectrons cause a cascade of ionization events, generating electron–hole pairs. The intensity of the generated current pulse is proportional to the energy of the incident γ or X-ray radiation. The potential distribution is created by surface electrodes and bias voltages to small collecting anode[15]. However, the energy range is typically limited to below 20–30 keV[16].

The energy resolution of detectors that use Metallic Magnetic Calorimeter (MMC) surpasses that of all previously mentioned technologies. An energy resolution of 10 eV Full Width at Half Maximum (FWHM) has been demonstrated at the 60 keV γ-line of an $^{241}$Am source[17,18]. A single MMC consists of a paramagnetic temperature sensor and a particle absorber in close thermal contact, which are together weakly connected to a thermal bath at 20 mK. When energy is deposited by an incident γ- or X-ray, the temperature of the calorimeter increases and the temperature-dependent magnetization of the sensor changes. The resulting change in magnetization of the sensor, which is located in a small magnetic field, changes. This can be precisely measured using a highly sensitive Superconducting Quantum Interference Device (SQUID) magnetometer.

In this study, an 89 kBq $^{225}$Ac sample is investigated, to the best of our knowledge, for the first time with an MMC-based detector[19–21]. For this purpose, calibration recordings are carried out and a suitable method for analyzing the data is developed. In addition to the already possible visualization of γ-lines of $^{221}$Fr and $^{213}$Bi with previously used detectors, this study clearly identifies $^{225}$Ac and $^{209}$Tl in the spectrum. Characteristic X-rays of $^{221}$Fr, $^{217}$At, $^{213}$Bi, $^{213}$Po, and $^{209}$Pb are also observed in the experimental data. This is, to the best of our knowledge, a first step towards qualitative and quantitative detection of all radionuclides of the decay chain of $^{225}$Ac, which is highly desirable for its application in nuclear medicine.

## Methods

A X-ray and γ-spectrum of an $^{225}$Ac sample and in addition spectra of $^{55}$Fe, $^{133}$Ba, and $^{241}$Am sources were measured and analyzed. An MMC detector was applied for the recordings[17].

### Material and set-up

The $^{225}$Ac stock was obtained from Solumedics (Obernetfelden, Switzerland), containing pure $^{225}$AcCl$_3$. The stock solution (89 kBq) was filled in an Eppendorf tube, sealed with Epoxy and placed into a second sealed Eppendorf tube as a second containment (photo of set-up in Fig. S1). To obtain the ultra-high-resolution spectrum, the $^{225}$Ac sample, which was in secular

equilibrium with its daughter radionuclides, was placed at a distance of ~8 cm in front of the MMC detector (with a gold absorber with an area of 16 mm²; thickness: 20 μm) for a period of 48.1 h. Figure S2 shows the activity profiles of the sample. The detector was located in a side arm of a dilution refrigerator. Additionally, a $^{55}$Fe source was present to identify gain drifts of the detector. For comparison, an SDD (Amptek, FAST SDD®, 70 mm² active area, 12.5 μm Be-window, Bedford) was placed at a distance of a few cm from the sample and collected data for 4 h. For calibration, spectra of $^{55}$Fe, $^{133}$Ba, and $^{241}$Am were measured for 32 h, of which 10 h were before and 21 h were after the $^{225}$Ac detection. A spectrum with a pure $^{55}$Fe sample was recorded for 9 h to exclude any potential influence of the $^{55}$Fe source during the $^{225}$Ac recording.

### Analysis of data

The acquired signal traces from the detector were saved with a data acquisition software (PAQS[22]). The saved traces were fitted based on a whitened matched filter, pileup- and $\chi^2$-filtered, corrected for temperature and gain drifts, and energy calibrated with a development version of a Python library (fitfiles)[23]. The $^{225}$Ac spectrum is energy calibrated with the eight most prominent lines from the calibration sources from 5.9 to 81 keV (MnK$_\alpha$, MnK$_\beta$, NpL$_{\beta1}$, AmL$_{\gamma2}$, CsK$_{\alpha2}$, CsK$_{\alpha1}$, Am$_{\gamma1}$, Ba$_\gamma$). The spectral analysis was performed with Origin. By comparing the three spectra, the $^{55}$Fe γ-lines from the calibration data were subtracted from the recorded $^{225}$Ac, $^{55}$Fe spectrum. The theoretical γ-lines and their transition probabilities for $^{225}$Ac and its daughter radionuclides ($^{221}$Fr, $^{217}$Rn, $^{217}$At, $^{213}$Po, $^{211}$Po, $^{213}$Bi, $^{211}$Bi, $^{209}$Tl, $^{207}$Tl and $^{209}$Pb) were obtained from a radionuclide database (radionuclide converter, JEFF-3.1, Nucleonica)[24,25].

The theoretical efficiency of the MMC was calculated in the range of 10–125 keV based on the absorber thickness of 20 μm[26]. Figure S3 shows this

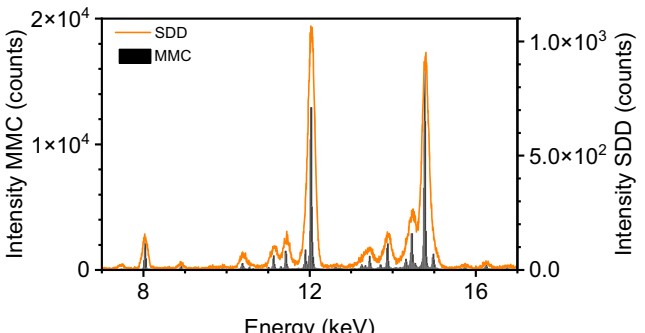

**Fig. 2 | Energy resolution comparison between SDD and MMC detector recorded spectra.** The spectra of the $^{55}$Fe, $^{225}$Ac sample recorded with an SDD (orange) and an MMC detector (black) demonstrate a significant difference in energy resolution. The FWHM at 5.9 keV is ~125 eV for the SDD, compared to 23 eV for the MMC.

efficiency curve. The theoretical γ-spectrum was compared with the detected histogram, and the individual γ-lines were assigned to their respective radioisotopes. Additionally, the energies of the L$_{\alpha1}$ and L$_{\beta1}$ characteristic X-ray fluorescence lines of $^{225}$Ac and its daughter radio-nuclides were determined, taking into account their emission probabilities, using the Hephaestus program within the Demeter suite[27] and PyMca[28], a collection of Python tools. This methodology enabled the assignement of peaks in the experimental spectra.

### Statistics and reproducibility

During the detection of the 89 kBq $^{225}$Ac sample with the MMC array, a total of 470,000 events was collected over a period of 48 h. In order to verify reproducibility, the spectra were compared at different measurement times to reveal statistical deviations. The spectra show no differences in the positions of signals at different measurement times. In addition, the signal-to-noise ratio is sufficient for all evaluated peaks in all energy ranges.

## Results

### Energy resolution of the MMC-based detector

First, the energy resolution of the MMC detector is investigated. For this purpose, a spectrum of the $^{55}$Fe$^{225}$Ac sample recorded with an SDD (orange) is compared with a spectrum measured using the MMC detector (black) (Fig. 2).

Despite the SDD having much better energy resolution than a standard γ-counter and its peak FWHM being one order of magnitude smaller than that of HPGe detectors[7], a clear improvement in resolution is observed in the MMC spectrum. An illustrative comparison can be made in the energy range between 11.6 and 12.3 keV. Here, the SDD shows only one, whereas the MMC detector reveals two distinct features at 11.9 keV and 12.0 keV, separated by 30 eV (Fig. S4). Additionally, in the energy range between 14 keV and 15 keV (Fig. S5), two overlapping peaks are observed with the SDD, whereas the MMC detector clearly resolves four distinct peaks at 14.3, 14.5, 14.8, and 15.0 keV. This clearly illustrates the superior energy resolution of the MMC detector compared to the SDD, allowing precise attribution of intensities at specific energy positions to $^{225}$Ac and its daughter radionuclides. In this experiment, the individual MMC pixels have a median FWHM energy resolution of 23 eV @ 6 keV and 61 eV @ 60 keV with an individual calibration uncertainty of about 1 eV.

### Calibration of the MMC detector

To be able to correct for time-dependent gain drifts of the detector response, the recording of a $^{55}$Fe source with its well-known lines is taken alongside the $^{225}$Ac sample. Additionally, a spectrum of $^{55}$Fe$^{241}$Am$^{133}$Ba is recorded with the MMC detector to calibrate the $^{225}$Ac and $^{55}$Fe spectra and identify the $^{55}$Fe peaks. Figure 3, panel (a) shows an overview histogram of the total

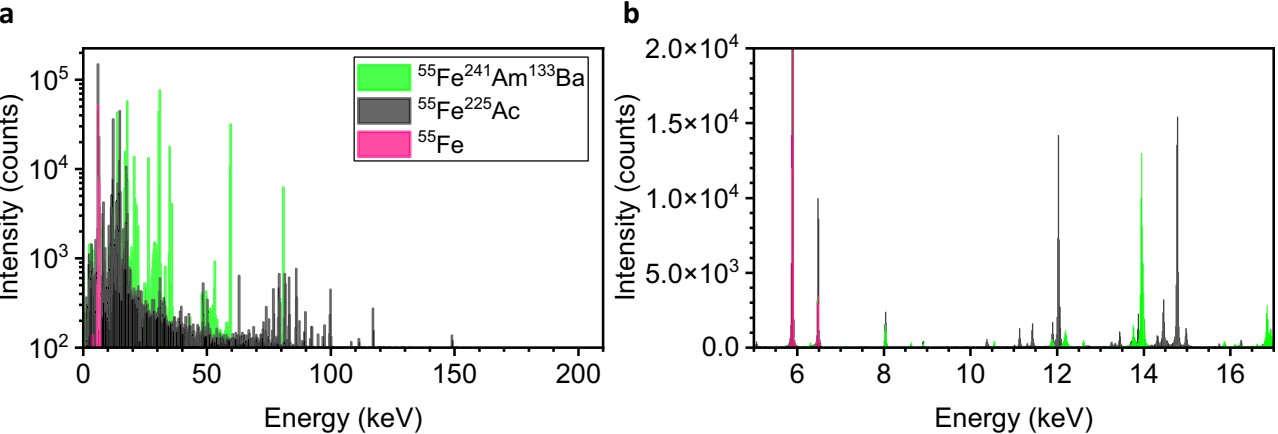

**Fig. 3 | Energy calibration spectra using several radionuclides.** Recorded γ- and X-ray spectra of $^{225}$Ac, $^{55}$Fe sample (black), $^{55}$Fe, $^{241}$Am, and $^{133}$Ba sample (green) and $^{55}$Fe as calibration data (pink) in energy regions of 0–200 keV (**a**) and 5–17 keV (**b**).

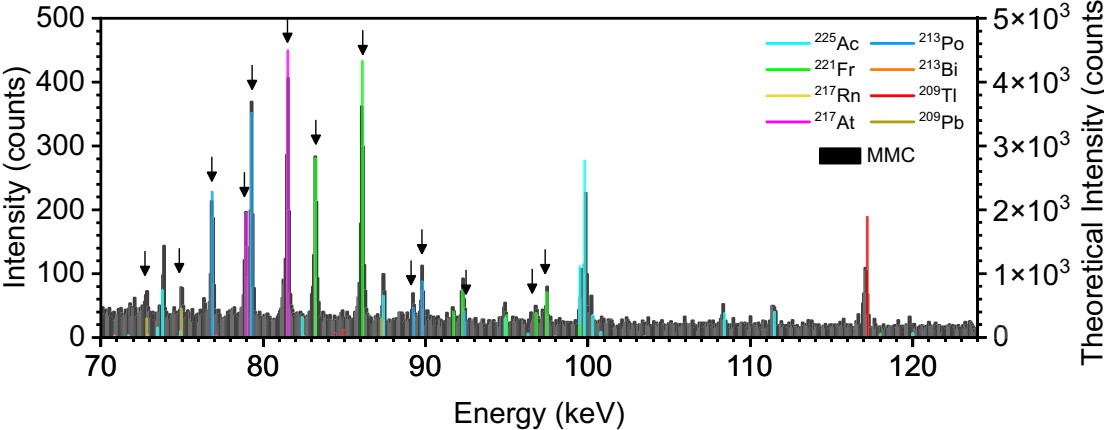

**Fig. 4 | γ-and X-ray spectrum of ²²⁵Ac and its daughter nuclides.** Recorded γ- and X-ray spectra of ²²⁵Ac sample (black, left y-axis) and theoretically calculated γ- and X-ray spectra of ²²⁵Ac and its dautghter nuclides (Nucleonica[24,29], Hephaestus[27,28]) with the efficiency of MMC detectors being taken into account (colored, right y-axis) (binsize:70 eV). The $K_\alpha$-X-rays of the respective isotopes are marked with arrows.

counts in the range 0-200 keV, while panel (b) is a close-up of the energy region between 5 and 17 keV.

The energy peaks of the calibration sample are observed in the low-energy region, with the most intense peak at 5.89 keV. No significant signals from ⁵⁵Fe are detected above 7 keV; therefore, all signals above this energy can be attributed to ²²⁵Ac and its daughter radionuclides. Figures S6–S12 show the experimental MMC spectrum of ²²⁵Ac and ⁵⁵Fe across different energy regions.

**Detected isotopes of ²²⁵Ac and its daughter radionuclides**

It is possible to assign gamma-lines of ²²⁵Ac and its daughters to specific signals using information from the calibration procedure and simulated spectrum of a 89 kBq ²²⁵Ac sample (radionuclide converter, JEFF-3.1, Nucleonica)[25]. This simulated spectrum is then corrected by the calculated efficiency of the MMC detectors and is visible as colored bars together with the recorded sample spectrum in the energy region of $E = 70–125$ keV in Fig. 4. The comparison is in logarithmic scale is plotted in Fig S13. The experimental spectrum is well-defined, shows minimal overlap of individual signals and has a low background. The theoretically predicted data matches the detected signals not only in energy positions but also in the relative intensities of the peaks as shown in Fig. 4. In Table 1 the significant peak are compared with theoretical data. As a result, in this energy range ($E = 70-125$ keV), the individually separated signals can be clearly assigned to the decay of the following radioisotopes: ²²⁵Ac, ²²¹Fr, ²¹³Bi, and ²⁰⁹Tl. Their γ-lines and X-rays are fairly intense and appear as isolated signals. ²²⁵Ac provides a distinct signal pattern, including several lines and therefore a fingerprint in the spectrum. Attention: $K_\alpha$-X-rays from ²²¹Fr occur during the decay of ²²⁵Ac to ²²¹Fr and are therefore characteristic signals for actinium presence. Of particular note are the two $K_\alpha$-lines at 79.29 keV and at 78.95 keV, which occur during the decay of ²¹³Bi and ²²¹Fr, which are resolved separately despite the small energy difference ($\Delta E < 0.5$ keV). In addition, several characteristic peaks of ²²⁵Ac can be resolved within the energy region around 100 keV, as well as individual sharp peaks that stand out prominently for the decay of ²²¹Fr (81.52 keV) and ²¹³Bi (76.86 keV). The signal of ²⁰⁹Tl at 117.2 keV can be identified as characteristic for the detection of ²²⁵Ac. The relative intensity ratio of experimental and calculated spectra is also of the same order of magnitude. However, there is still room for improvement with regards to the detector efficiency calculation, as the theoretical intensities in the energy range of 70-80 keV are lower than those determined experimentally.

There are several reports of ²²⁵Ac spectra examined with different types of detectors; for example a NaI detector was applied by Usmani et al.[4]. In comparison to our spectrum, the authors recorded a much higher background which makes weak peaks indifferentiable. Furthermore, due to low

energy resolution and low sensitivity in the low energy region, no individual signals could be assigned to ²²⁵Ac and its daughters. Instead, they show three broad main peaks (78, 218, and 440 keV) which include several γ-lines of all radionuclide daughters of ²²⁵Ac. In comparison to a NaI detector, a Canberra high-resolution γ-spectrometer based on an HPGe detector achieves an energy resolution of 1.5 keV. This type of detector was applied by Apostolidis et al.[12]. However, the signals at E < 100 keV were still overlapping in comparison to the detected spectra using the MMC detectors in our study. Bertuccio et al. reached an energy resolution of 124 eV @ 5.9 keV (⁵⁵Fe line) with an SDD. However, up to now no ²²⁵Ac spectra are recorded, possibly because the accessible energy range is relatively limited[29] and the detectable nuclides are restricted. The MMC detectors offer a high energy resolution across a very broad energy range. As a result, it is possible to obtain sharp signals of ²²⁵Ac and to detect additional daughter nuclides beyond ²²¹Fr and ²¹³Bi, even below 100 keV. In particular, intensities of ²²⁵Ac, ²²¹Fr, ²¹³Bi, and ²⁰⁹Tl are assignable and can therefore clearly be detected.

The high energy resolution (FWHM 23 eV @ 5.9 keV) and the comparison of the recorded spectrum with the theoretical spectrum of ²²⁵Ac and its daughter radionuclides enables the clear assignment of radionuclides to spectral signals. In Fig. 3 a substantial portion of the detected signals in the low energy region can be assigned to ⁵⁵Fe. Other signals, such as those at 11.1 and 11.4 keV, are also detected, which do not corespond to ⁵⁵Fe.

The spectrum in Fig. 5 shows the experimental spectrum of the ²²⁵Ac sample in the energy range of 9.5 keV to 17.5 keV (in Fig. S14, comparison on a logarithmic scale). In this energy region, no γ-lines of ²²⁵Ac and its daughter radionuclides are expected. However, several features are detected in this energy area. These are due to emission of characteristic X-rays from the different daughter nuclides of ²²⁵Ac. The decay leaves the atom in an excited state. To reach a ground state it—amongst others—emits characteristic fluorescence. The $L_\alpha$ emission of ²²⁵Ac and its daughters is located in the energy range from 10.3 to 12.6 keV, while the $L_\beta$ X-ray emission lines are found between 12.21 and 15.71 keV. Their theoretical values are plotted in different colors in Fig. 5[27,28]. The theoretical intensities are scaled by the activity of the nuclide as well as a factor of 7 for Ac, 5.5 for Bi, 20 for Pb and Po, 500 for Fr, and 80 for At, to match the highest experimentally observed peak. In particular, the radionuclides ²²¹Fr, ²¹⁷At, ²¹³Bi, ²¹³Po, and ²⁰⁹Pb exhibit characteristic X-ray emission peaks at the energy positions detected by the MMC detector. This clear assignment holds the promise of future quantification of additional nuclides in the decay chain. $L_\alpha$- and $L_\beta$- signals from the undecayed ²²⁵Ac mother nuclide are also potentially recognizable but fairly weak. This may be explained by the effect of Particle Induced X-ray Emission (PIXE) or, less probable, because of gamma-induced L-edge excitation by weak gamma emission lines over the entire decay chain. PIXE

**Table 1 | Comparison of significant peaks for X-Ray and γ-spectra of $^{225}$Ac determined with MMC detector experimentally and theoretical data from Nucleonica, Hephaestus and PyMca[24,27–29,37]**

| Experimental MMC Data | | | Theoretical Data | | | Experimental MMC Spectra | | | Theoretical Data | | |
|---|---|---|---|---|---|---|---|---|---|---|---|
| Peak Name | Energy Position [keV] | Error [keV] | Peak Name | Energy Position [keV] | Error [keV] | Peak Name | Energy Position [keV] | Error [keV] | Peak Name | Energy Position [keV] | Error [keV] |
| $^{213}$Po | 9.655 | 0.002 | $^{213}$Po | 9.658 | 0.007[26,27] | $^{209}$Pb* | 72.82 | 0.018 | $^{209}$Tl | 72.804 | 0.001[26,27] |
| $^{217}$At | 9.9 | 0.002 | $^{217}$At | 9.897 | 0.001[26,27] | $^{225}$Ac | 73.87 | 0.003 | $^{225}$Ac | 73.86 | 0.02[36] |
| $^{221}$Fr | 10.383 | 0.001 | $^{221}$Fr | 10.381 | 0.002[26,27] | $^{209}$Pb* | 74.98 | 0.006 | $^{209}$Tl | 74.969 | 0.001[26,27] |
| $^{209}$Pb | 10.551 | 0.001 | $^{209}$Pb | 10.551 | 0.001[26,27] | $^{213}$Po* | 76.86 | 0.004 | $^{213}$Bi | 76.862 | 0.002[26,27] |
| $^{213}$Bi | 10.842 | 0.003 | $^{213}$Bi | 10.839 | 0.001[26,27] | $^{217}$At* | 78.95 | 0.004 | $^{221}$Fr | 78.95 | 0.003[26,27] |
| $^{213}$Po | 11.017 | 0.001 | $^{213}$Po | 11.016 | 0.001[26,27] | $^{213}$Po* | 79.29 | 0.005 | $^{213}$Bi | 79.29 | 0.001[26,27] |
| $^{213}$Po | 11.133 | 0.0005 | $^{213}$Po | 11.13 | 0.001[26,27] | $^{217}$At* | 81.51 | 0.003 | $^{221}$Fr | 81.52 | 0.001[26,27] |
| $^{217}$At | 11.309 | 0.0004 | $^{217}$At | 11.306 | 0.001[26,27] | $^{221}$Fr* | 83.22 | 0.004 | $^{225}$Ac | 83.23 | 0.001[26,27] |
| $^{217}$At | 11.428 | 0.0005 | $^{217}$At | 11.426 | 0.001[26,27] | $^{209}$Tl | 85.21 | 0.029 | $^{209}$Tl | 84.936 | 0.001 |
| $^{221}$Fr | 11.900 | 0.0006 | $^{221}$Fr | 11.896 | 0.001[26,27] | $^{221}$Fr* | 86.10 | 0.003 | $^{225}$Ac | 86.10 | 0.001[26,27] |
| $^{221}$Fr | 12.033 | 0.0005 | $^{221}$Fr | 12.031 | 0.001[26,27] | $^{225}$Ac | 87.42 | 0.004 | $^{225}$Ac | 87.41 | 0.02[36] |
| $^{209}$Pb | 12.619 | 0.002 | $^{209}$Pb | 12.6144 | 0.001[26,27] | $^{213}$Po* | 89.24 | 0.011 | $^{213}$Bi | 89.25 | 0.005[26,27] |
| $^{221}$Fr | 13.255 | 0.0008 | $^{221}$Fr | 13.255 | 0.001[26,27] | $^{213}$Po* | 89.79 | 0.004 | $^{213}$Bi | 89.8 | 0.004[26,27] |
| $^{213}$Po | 13.341 | 0.002 | $^{213}$Po | 13.341 | 0.014[26,27] | $^{221}$Fr | 91.71 | 0.003 | $^{221}$Fr | 91.72 | 0.001 |
| $^{213}$Po | 13.446 | 0.0001 | $^{213}$Po | 13.443 | 0.003[26,27] | $^{221}$Fr | 92.3 | 0.005 | $^{221}$Fr | 92.3 | 0.011[26,27] |
| $^{217}$At | 13.709 | 0.0004 | $^{217}$At | 13.708 | 0.013[26,27] | $^{221}$Fr/ $^{225}$Ac | 94.86 | 0.008 | $^{221}$Fr | 94.9 | 0.02[36] |
| $^{217}$At | 13.877 | 0.0002 | $^{217}$At | 13.875 | 0.001[26,27] | $^{221}$Fr* | 96.79 | 0.007 | $^{225}$Ac | 96.81 | 0.005[26,27] |
| $^{217}$At | 14.077 | 0.001 | $^{217}$At | 14.073 | 0.006[26,27] | $^{221}$Fr* | 97.45 | 0.005 | $^{225}$Ac | 97.47 | 0.001[26,27] |
| $^{217}$At | 14.178 | 0.002 | $^{217}$At | 14.166 | 0.008[26,27] | $^{225}$Ac | 99.59 | 0.010 | $^{225}$Ac | 99.65 | 0.004[36] |
| $^{221}$Fr | 14.319 | 0.0006 | $^{221}$Fr | 14.319 | 0.007[26,27] | $^{225}$Ac | 99.86 | 0.004 | $^{225}$Ac | 99.9 | 0.004[36] |
| $^{221}$Fr | 14.460 | 0.0005 | $^{221}$Fr | 14.458 | 0.015[26,27] | $^{225}$Ac/ $^{221}$Fr | 100.2 | 0.013 | $^{225}$Ac/ $^{221}$Fr | 100.9/ 100.25 | 0.03[36] |
| $^{221}$Fr | 14.771 | 0.0003 | $^{221}$Fr | 14.770 | 0.001[26,27] | $^{225}$Ac | 108.31 | 0.008 | $^{225}$Ac | 108.4 | 0.02[36] |
| $^{221}$Fr | 14.976 | 0.0004 | $^{221}$Fr | 14.978 | 0.008[26,27] | $^{225}$Ac | 111.42 | 0.014 | $^{225}$Ac | 111.53 | 0.02[36] |
| $^{213}$Po | 15.743 | 0.0006 | $^{213}$Po | 15.742 | 0.002[26,27] | $^{209}$Tl | 117.11 | 0.010 | $^{209}$Tl | 117.2 | 0.001 |
| $^{217}$At | 16.249 | 0.001 | $^{217}$At | 16.249 | 0.003[26,27] | | | | | | |
| $^{221}$Fr/ $^{217}$At | 16.743 | 0.0008 | | | 0.022[26,27] | | | | | | |
| $^{221}$Fr | 17.303 | 0.0004 | $^{221}$Fr | 17.302 | 0.002[26,27] | | | | | | |

Stars (*) describe $K_\alpha$-X-rays from the respective isotopes.

describes a process in which high-energy charged particles, such as alpha particles, with an energy of several MeV induce X-ray emissions in surrounding atoms. Usually this is observed when alpha particles from, for example, a $^{244}$Cm source irradiates the material to be analyzed[30,31]. PIXE can be an explanation for the appearance of unexpected X-ray lines of undecayed actinium in the MMC spectrum (12.65 and 15.71 keV). Previous PIXE spectra mainly describe this phenomenon for light elements like magnesium, aluminum and silicon by irradiation with 3-5 MeV energy helium ion beams produced by $^{210}$Po or $^{244}$Cm sources[31]. Heirwegh et al.[32] and Cureatz et al.[31] investigate Mg, Al, and Si samples, while Pogrebnjak et al.[33] treats Fe, Cu and Zn samples. According to Konya et al.[34], trace concentrations of elements from atomic number 13 onward can be detected. For K-lines, the most sensitive range is $20 < Z < 35$ and for L-lines $75 < Z < 85$[34], rationalizing the potential detection of $L_\alpha$- and $L_\beta$-lines for Ac ($Z = 89$) in our spectrum. Kurosawa et al. propose that elements from sodium ($Z = 11$) can be detected with PIXE at a few ppm per sample (Ryan et al.)[35]. Keizo Ishii et al., describe possible PIXE analysis from sodium up to uranium ($Z = 92$) at an energy resolution of 220 eV[30].

The existing PIXE recordings are limited by their detector technologies due to limitations in detector efficiency, low energy resolution and self-absorption in the sample. By combining improved detector efficiency and energy resolution in a broad energy range, MMC detectors may access PIXE experiments beyond the previous boundaries. The PIXE effect has been investigated in the past, but primarily for lighter elements than the $^{225}$Ac investigated in this work due to limited detector technologies. Therefore, finding of the PIXE effect within the $^{225}$Ac sample seems at least plausible.

## Discussion

This study demonstrates the remarkable capabilities of using an MMC detector to record $^{225}$Ac and daughters' X-ray and γ-spectra in a wide energy range of 5–120 keV with high energy resolution, sensitivity, and precise calibration, highlighting its potential to significantly advance both the analysis and detection of these radionuclides. The ultra-high energy resolution of the MMC detector yields a well defined $^{225}$Ac spectrum (FWHM 23 eV @ 5.9 keV and 61 eV @ 60 keV), revealing more lines than previously published spectra. Individual γ- and X-ray-lines occurring during decay of

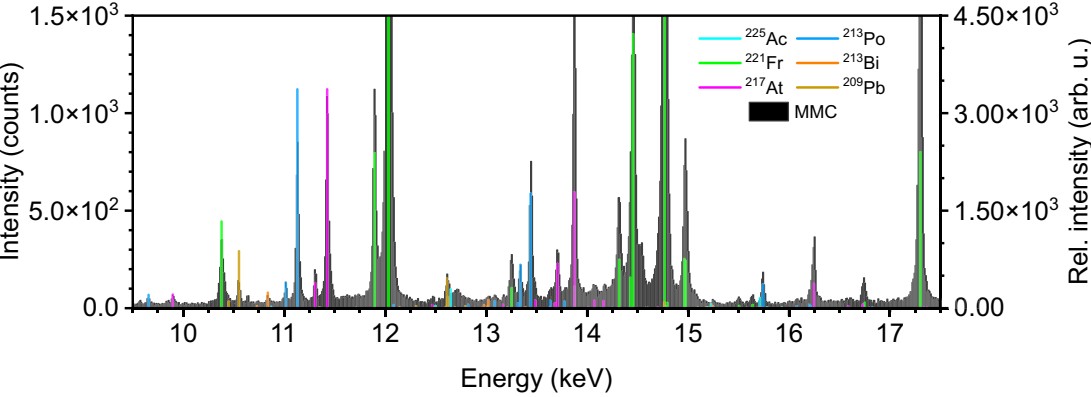

**Fig. 5 | X-ray spectrum of [225]Ac and its daughter nuclides.** Recorded spectra of [225]Ac sample ((MMC) black, left y-axis) and theoretical X-rays of [225]Ac and its daughters (colored, right y-axis (Hephaestus and PyMca[27,28])) are plotted (binsize: 10 eV).

[225]Ac, [221]Fr, [213]Bi, and [209]Tl are clearly resolved. Their relative intensities match theoretical predictions and provide a basis for future precise quantitative analysis. Therefore, the high resolution in lower energy ranges provides a level of detail previously unattainable. Moreover, the MMC detector enables access to additional phenomena, such as possible α-induced X-ray emissions from non-decayed [225]Ac, as well as characteristic X-ray emissions from the daughter nuclides [221]Fr, [217]At, [213]Bi, [213]Po, and [209]Pb in the energy range from 10.55 to 14.77 keV. In total, due to the ultra-high energy resolution, the peaks in the detected spectrum can be assigned to [225]Ac, [221]Fr, [213]Bi, [209]Tl, [217]At, [213]Po, and [209]Pb. This represents the entire decay chain except for [221]Ra and [217]Rn. This is, however, not surprising due to the low decay probabilities and half-lives of [221]Ra and [217]Rn which are -as a result- negligibly present in daughter nuclide mixtures.

With the demonstration that individual nuclides in the secular equilibrium of [225]Ac with its daughters can be reliably characterised with the MMCs high energy resolution, the application in quality assurance of radionuclide and radiopharmaceutical production is evident. Additionally, this selectivity is the foundation for potential in vivo imaging. Previously, various surrogates such as [68]Ga, [89]Zr, [177]Lu, [152]Tb, [132]La, and [134]La were proposed for the purpose of enabling [225]Ac imaging and dose distribution determination[37–39]. Due to chemical differences between these surrogates and [225]Ac, all of these surrogate imaging methods can only to some extend be used for the dose estimation. Also daughter nuclides cannot be surrogated with these methods. In addition, previous methods cannot take into account the phenomenon of [225]Ac being liberated from the chelator of the radiopharmaceutical, which leads to a falsification of the dose values[40].

Before direct [225]Ac imaging without surrogates for in vivo applications can be realized, further investigation of the absorption of X-rays in tissues is needed. These absorption processes depend on the patient's mass, the location of the radionuclide in the body, and the energy of the photons. Medical physics calculations are required before quantification of individual radionuclides in in vivo experiments is possible. To obtain a rough estimate, a simplified mouse model can be considered. Mice consist of ~70% water[41] and photon absorption in water varies with energy. Low-energy X-ray transmission through 1 mm and 10 mm of water is plotted against photon energy in Fig. S15 and Fig. S16. In the low-energy X-ray region around 14 keV (Fig. 5), transmissions of 82% (1 mm) and 13% (10 mm) are observed. Signals above background levels remain detectable, and absorption can be corrected for. Thus, for tissue sections 2–5 μm thick, entire organs, and potentially tumor tissue in mice, imaging using this region is possible. Bone and organ densities can reach up to 1.5 g/cm³, with corresponding absorption shown in Fig. S17 and Fig. S18. Beyond the centimeter scale, transmission decreases significantly for the low-energy X-rays. While this can be partially offset by improved detector sensitivity, it remains a limiting factor. However, this limitation loses significance for photons with energies of 70 keV and above. This corresponds to the second energy region (Fig. 4) investigated in this paper which achieve transmissions of 14% or more even through 100 mm of water. It has very recently been shown that photons in the 78 keV energy range can make important contributions to dosimetry of [221]Fr and [213]Bi in patients[4,42]. Future investigations will determine which energy regions in the spectrum are particularly useful for which application.

For future work, we propose the development of a detector for γ- and X-ray spectroscopy up to 120 keV or higher, optimized for applications in medical physics. Such a detector will feature thicker absorbers of ~100 – 200 μm. To maintain the, to the best of our knowledge, unprecedented energy resolution, the lateral size of the absorbers can be reduced accordingly, because the energy resolution is roughly proportional to the square root of the volume of the absorber. This would additionally lead to higher pixel density and thus higher spatial resolution. Furthermore, improved statistics from longer measurements can easily enhance the energy calibration and spectral analysis. Another direction of MMC detector developments could lead to ultra-high energy resolution, enabling the detection of chemical shifts. From these, it will be possible to infer chemical information about the changes in the speciation of the daughter nuclides due to bond breakage or cellular interaction. This cannot be obtained with any of the currently available detector technologies. Additionally, employing collimator optics in vivo will facilitate high spatial resolution, enhancing the precision and accuracy of in vivo experiments. The well-resolved spectra allow the easy detection and quantification of impurities and will further improve the reliability and quality of radiation-based experiments and treatments. With ongoing development, the MMC detectors can become invaluable tools for advancing both basic research and clinical applications in nuclear medicine and radiopharmaceuticals while advancing personalized TαT.

To conclude, using MMC detectors with high energy resolution significantly improves the identification of γ and X-ray emissions from [225]Ac and its daughter radionuclides, even in regions where weak signals are typically difficult to differentiate. The results highlight the capability of these detectors to resolve complex spectra, including characteristic fluorescence arising from the PIXE effect, offering precise characterization of radioactive materials. The results presented in this study form the basis for further work and show that this approach to distinguish individual nuclides using the high-energy resolution detector technology employed is fundamentally feasible.

## Data availability
Original experimental data of this study are available through Zenodo https://doi.org/10.5281/zenodo.17795035[23].

## Code availability

The code used for this study is released under the CC BY license and can be obtained through Zenodo https://doi.org/10.5281/zenodo.17795035[23].

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

## Acknowledgements

K.M. was supported by the Federal Ministry of Education and Research (BMBF) and the Baden-Württemberg Ministry of Science as part of the Excellence Strategy of the German Federal and State Governments as KIT future fields stage II project—InnoAlpha. D.U. was supported by High Resolution and High Rate Detectors in Nuclear and Particle Physics (HighRR) [DFG GRK 2058]. K.v.S. was supported by the ETH Research Grant 22-2 ETH-023.

## Author contributions

The manuscript was written through the contributions of all authors. All authors have given approval to the final version of the manuscript. Kiara Maurer: Data curation, Investigation, Formal analysis, Visualization, Validation, Writing – original draft, Writing – review & editing. Daniel Unger: Investigation, Formal analysis, Data curation, Methodology, Software, Validation, Writing – review & editing. Daniel Hengstler: Methodology, Software, Writing – review & editing. Martin Behe: Resources, Writing – review & editing. Andreas Knecht: Project administration, Writing – review & editing. Katharina von Schoeler: Investigation, Formal analysis, Writing – review & editing. Tonya Vitova: Funding acquisition, Writing – review & editing. Martina Benesova-Schäfer: Funding acquisition, Writing – review & editing. Andreas Fleischmann: Project administration, Writing – review & editing. Loredana Gastaldo: Funding acquisition, Supervision, Writing – review & editing. Carmen Wängler: Project administration, Writing – review & editing. Christian Enss: Project administration, Writing – review & editing. Bianca Schacherl: Conceptualization, Funding acquisition, Project administration, Investigation, Supervision, Writing – review & editing.

## Funding

## Competing interests

The authors declare no competing interests.
