## [Transparent Peer Review file · Communications Medicine]

Advancing towards cancer theragnostic by probing the ^{225}Ac decay chain with ultra-high-resolution metallic magnetic calorimeter based detectors

Corresponding Author: Dr Bianca Schacherl

Version 0:

Reviewer comments:

Reviewer #1

(Remarks to the Author)

The present work is dedicated to the adaptation of a detector based on a metal magnetic calorimeter (MMC) for measuring the gamma spectrum in the range of 5-125 keV. It is evident that, given the high resolution of this method (several eV), this work is of great scientific interest for the study of atomic-nuclear processes. Nevertheless, it is this reviewer's opinion that the work is somewhat incomplete.

Firstly, the authors of the text make mention of the fact that there is a high degree of concordance between the calculated and experimental data. However, for methodological work, it is necessary to present the experimental results obtained by the authors in tabular form, indicating the uncertainties for specific gamma transitions of different nuclides in the ^{225}Ac decay chain, and to compare them with the available data from the Table of Isotopes and their errors.

Secondly, the authors have not demonstrated the reliability of this method for determining dose distribution in various organs during targeted alpha therapy. It is important to note that no other characteristics apart from the aforementioned 'very good energy resolution' have been specified. Consequently, the registration efficiency and the energy dependence of the registration efficiency are as yet unknown. Furthermore, there is no discussion of how the self-absorption of X-rays, which is proposed to be used to determine the dose distribution in various organs of patients, including those of different masses, will be taken into account, and how accurate these dose estimates will be, taking this factor into account. At present, well-established paired systems (simultaneous injection of an alpha-emitting radionuclide with ^{68}Ga or ^{152}Tb) are utilised to visualise the applied dose on PET/SPECT. The article does not provide sufficient evidence to support the hypothesis that the utilisation of characteristic radiation accompanying the decay of ^{225}Ac for the purpose of dose visualisation is valid.

In addition, a brief remark is warranted concerning the authors' assertions regarding the efficacy of registering semiconductor spectrometers. It is important to note that modern semiconductor spectrometry, based on ultra-pure germanium detectors, has a registration efficiency of up to 50% compared to $3''\times 3''$ NaI(Tl) detectors, rather than 10%.

The work can be concluded to be of scientific interest and to demonstrate a high level of methodical approach, with a niveau of execution that merits publication in a scientific journal. However, in my opinion, it is evident that the work is not suitable for publication in the journal "Communications Medicine", since the study and refinement of separate X-rays and low-intensity gamma transitions of radioisotopes used in target alpha therapy is not a relevant task for this type of therapy. The properties of these radioisotopes are well researched. This article studies low-energy gamma transitions and X-ray radiation. These do not have a significant additional effect on the dose administered to the patient. Notwithstanding, the method itself is of significant scientific interest for fundamental nuclear physics and is more relevant to the subject matter of the journal Nuclear Instruments and Methods in Physics A.

Reviewer #2

(Remarks to the Author)

General comments:

The manuscript describes gamma and x-ray spectrometric analysis of ^{225}Ac and its decay chain with high-resolution cryogenic MMC detector technique. The applied experimental setting allows high resolution and quantification of relevant radionuclide mixture.

Ac-225 is one of the most essential alpha emitters for targeted radionuclide treatment of cancer. Improvement of quantitative in vivo imaging of Ac-225-radiolabeled compound along with the organ specific dosimetry would be significant achievement in the field.

Although, the results suggest the high potential of MMC for gamma and x-ray spectrometric analysis of Ac-225, the paper doesn't provide strong storyline with enlightenment of technical realization of the technique for in vivo imaging. Experimental set-up and investigated sample is not necessarily representative and justify the conclusion; limitations are not properly addressed. Significant part of the study is focused on analysis of very low energy photons, which are not relevant for in vivo imaging. Therefore, the manuscript should be improved before publishing.

Specific comments:

The title is not precise; Term "theragnostic" includes diagnostic component and doesn't reflect the problem statement;

Abstract and later – Term "gamma-spectroscopy" might be more specific, since the paper refer to the practical application of the selected technique;

Introduction

Introduction should focus on the state-of-the-art technology, relevant for the storyline (i.e. SPECT); it is not clear, why it includes basic discussion on PET imaging or HPGe detector.

Line 79 – SPECT scintillators such as NaI or CsI are considered to have high sensitivity;

Line 80 – the statement is not supported by Reference #3. It might be supported by Reference #10(?);

SDDs is included in the experimental setting as benchmark but is not properly introduced.

Statement lines 102 – 105 is confusing and seems to be not relevant. It is as well not part of the subsequent discussion.

MMC introduction doesn't include information on material of the absorber and its dimension. Introduction and later discussion should properly address potential limitations. In particularly please discuss sensitivity and limitations related to the size of the absorber. Potential technical limitations/ advantages such as time resolution and compatibility of MMC with imaging algorithm, how spatial resolution could be achieved.

Results

Results and later discussion are dominated by observations made in very low energy spectrum. Even the results seem to be encouraging, those low energy gamma and X-rays, will not contribute to in vivo imaging. Additional observations and other potential areas of research should be clearly differentiated from the main storyline.

Discussion

Please consider comments above. Properly address all limitations. Provide outlook for further development and implementation for in vivo imaging, and other areas of research. Are follow-up experiments planed (with representative samples, collimator technique, etc)?

Version 1:

Reviewer comments:

Reviewer #1

(Remarks to the Author)

As a reviewer, I am satisfied with the responses to my comments and the edits made to the article. It is my opinion that the article in question has been rendered more relevant to the subject matter of the journal in its second iteration.

Reviewer #2

(Remarks to the Author)

Answer to reviewers' comments

Reviewer #1 (Remarks to the Author):

Comment 1: The present work is dedicated to the adaptation of a detector based on a metal magnetic calorimeter (MMC) for measuring the gamma spectrum in the range of 5-125 keV. It is evident that, given the high resolution of this method (several eV), this work is of great scientific interest for the study of atomic-nuclear processes. Nevertheless, it is this reviewer's opinion that the work is somewhat incomplete. Firstly, the authors of the text make mention of the fact that there is a high degree of concordance between the calculated and experimental data. However, for methodological work, it is necessary to present the experimental results obtained by the authors in tabular form, indicating the uncertainties for specific gamma transitions of different nuclides in the ^{225}Ac decay chain, and to compare them with the available data from the Table of Isotopes and their errors.

Answer: We thank the reviewer; we also think this method is of great scientific interest for several fields. We have now added in the manuscript a table with the energies of different transitions including values which are derived by fitting the spectrum as well as literature data both with uncertainties is given in table (Table 1).

Comment 2: Secondly, the authors have not demonstrated the reliability of this method for determining dose distribution in various organs during targeted alpha therapy. It is important to note that no other characteristics apart from the aforementioned 'very good energy resolution' have been specified. Consequently, the registration efficiency and the energy dependence of the registration efficiency are as yet unknown. Furthermore, there is no discussion of how the self-absorption of X-rays, which is proposed to be used to determine the dose distribution in various organs of patients, including those of different masses, will be taken into account, and how accurate these dose estimates will be, taking this factor into account.

Answer: We thank the reviewer for this absolutely justified comment. In fact, we have not shown here that the new detector technology is actually suitable to determine the organ and tumor doses from the radiation emitted by ^{225}Ac and its daughters *in vivo*. That is due to the fact that we initially intended to use these measurements to investigate whether the MMC detectors are in principle suitable for this application as this has not been studied before. In particular, we intended to determine whether this technology enables the individual nuclides in the secular equilibrium of ^{225}Ac with its daughters to be reliably distinguished with sufficiently high energy resolution. We were able to demonstrate impressively that this is the case. Of course, this is only the beginning towards actual *in vivo* imaging of ^{225}Ac and its daughters in living organisms, especially with regard to the attenuation of the corresponding low-energy photons mentioned by the reviewer and the corresponding reconstruction of the obtained images. As mentioned by the reviewer, this means that aspects such as photon absorption (which depends on the patient's total mass, the location of the nuclides in the body, the energy of the photons, etc.) also need to be examined using medical physics calculations before quantifiable results can be obtained. However, this will be the subject of further work. The results presented here provide the basis for this further work and show that this approach to differentiate between individual nuclides using the high energy resolution detector technology employed is fundamentally viable. We have firstly included a photon absorption efficiency curve for the currently used gold absorber in SI4 and secondly included this more differentiated discussion in the discussion section:

"Before direct ^{225}Ac imaging without surrogates for *in vivo* applications can be realized further investigation of the absorption of X-rays in tissues is needed. These absorption processes depend on the patient's mass, the location of the radionuclide in the body, and the energy of the photons. There is a need of medical physics calculations before quantification in *in vivo* experiments of the

individual radionuclides are possible. To obtain a rough estimate, a simplified mouse model can be considered. Mice consist of approximately 70% water⁴¹ and photon absorption in water varies with energy. Low energy X-ray transmission through 1 mm and 10 mm of water is plotted against photon energy in Figure S15 and Figure S16. In the low energy X-ray region around 14 keV (Figure 5), transmissions of 82% (1 mm) and 13% (10 mm) are observed. Signals above background levels remain detectable, and absorption can be corrected for. Thus, for tissue sections 2–5 μm thick, entire organs, and potentially tumor tissue in mice imaging using this region is possible. Bone and organ densities can reach up to 1.5 g/cm^3 , with corresponding absorption shown in Figure S17 and Figure S18. Beyond the centimeter scale, transmission decreases significantly for the low energy X-rays. While this can be partially offset by improved detector sensitivity, it remains a limiting factor. However, this limitation loses significance for photons with energies of 70 keV and above. This corresponds to the second energy region (Figure 4) investigated in this paper which achieve transmissions of 14% or more even through 100 mm of water. It has very recently been shown that photons in the 78 keV energy range can make important contributions to dosimetry of ²²¹Fr and ²¹³Bi in patients^{4,42}. Future investigations will determine which energy regions in the spectrum are particularly useful for which application.”

Comment 3: At present, well-established paired systems (simultaneous injection of an alpha-emitting radionuclide with ⁶⁸Ga or ¹⁵²Tb) are utilised to visualise the applied dose on PET/SPECT. The article does not provide sufficient evidence to support the hypothesis that the utilisation of characteristic radiation accompanying the decay of ²²⁵Ac for the purpose of dose visualisation is valid.

Answer: The reviewer is indeed right in saying that various approaches are currently being pursued to estimate the dose from ²²⁵Ac and its daughters in the various organs *in vivo*. Various surrogates such as ⁸⁹Zr, ¹⁷⁷Lu, ¹³²La, and ¹³⁴La were proposed for this purpose, but all of these surrogate imaging methods suffer from the fact that they can to some extent be used to estimate the dose from ²²⁵Ac (albeit not directly and therefore not accurately due to chemical differences of the different nuclides and thus potentially also different pharmacokinetic properties of the compounds), but not the dose caused by the daughters. Further, these surrogates cannot depict phenomena such as liberation of ²²⁵Ac from the applied radiopharmaceutical, resulting in falsification of the dosimetry results (DOI: 10.1088/1361-6560/ac5fe0). This currently results in considerable deviations in the dose determined by clinical surrogate imaging in the tumor and also in organs, which is why it is necessary to identify approaches to actually detect all relevant nuclides directly by imaging. Currently, this is not possible. The best approach currently available is limited to the visualization of ²²¹Fr and ²¹³Bi and excludes ²²⁵Ac (DOIs: 10.1097/RLU.0000000000002525 and 10.1007/s00259-024-06681-2). The approach we present here could be a solution to this problem, although further research is of course needed to achieve this goal (see above).

Comment 4: In addition, a brief remark is warranted concerning the authors' assertions regarding the efficacy of registering semiconductor spectrometers. It is important to note that modern semiconductor spectrometry, based on ultra-pure germanium detectors, has a registration efficiency of up to 50% compared to 3”x3” NaI(Tl) detectors, rather than 10%.

Answer: Yes, we agree with the reviewers comments completely. 3”x3” NaI (Tl) detectors reach an efficiency of 22.6% at 1274.5 keV with a distance of 2cm in experimental studies (DOI: 10.31590/ejosat.443565). The 50% relative efficiency then corresponds to a total efficiency of about 10% (DOI: 10.4103/rpe.RPE_1_19).

Comment 5: The work can be concluded to be of scientific interest and to demonstrate a high level of methodical approach, with a niveau of execution that merits publication in a scientific journal. However, in my opinion, it is evident that the work is not suitable for publication in the journal “Communications Medicine”, since the study and refinement of separate X-rays and low-intensity gamma transitions of radioisotopes used in target alpha therapy is not a relevant task for this type of therapy. The properties of these radioisotopes are well researched. This article studies

low-energy gamma transitions and X-ray radiation. These do not have a significant additional effect on the dose administered to the patient.

Answer: We thank the reviewer for this comment, but cannot agree to this statement. These low-energy photons are the basis for direct imaging and thus quantification of ^{225}Ac using SPECT. In particular, photons at 83 and 86 keV would be very well suited for this purpose and, as shown in the manuscript, can be reliably distinguished from photons of other nuclides. It has very recently been shown that photons in this energy range (specifically: 78 keV) can make important contributions to dosimetry of ^{221}Fr and ^{213}Bi in patients (DOIs: 10.1097/RLU.0000000000002525 and 10.1007/s00259-024-06681-2). We are therefore convinced that the results presented here will be of great interest for medical applications, even though further research is of course necessary before they can be used in patients. We have now stronger emphasized this aspect in the manuscript. An additional application is the use in quality assurance during production of medical isotopes and radiopharmaceuticals. This could be achieved with the current setup and has been voiced by several medical isotope producing companies already.

Comment 6: Notwithstanding, the method itself is of significant scientific interest for fundamental nuclear physics and is more relevant to the subject matter of the journal Nuclear Instruments and Methods in Physics A.

Answer: We thank the reviewer for seeing the significant scientific interest in the method. This method is indeed new to the field of medicine but with the arguments above think there is broad interest from several sides in the medical community. Therefore, we believe it fits perfectly into Communication Medicine to spearhead several applications of this emerging technique.

Reviewer 2:

Comment 1: The manuscript describes gamma and x-ray spectrometric analysis of Ac-225 and its decay chain with high-resolution cryogenic MMC detector technique. The applied experimental setting allows high resolution and quantification of relevant radionuclide mixture. Ac-225 is one of the most essential alpha emitters for targeted radionuclide treatment of cancer. Improvement of quantitative in vivo imaging of Ac-225-radiolabeled compound along with the organ specific dosimetry would be significant achievement in the field.

Answer: We would like to thank the reviewer for this analysis, which is consistent with our own view.

Comment 2: Although, the results suggest the high potential of MMC for gamma and x-ray spectrometric analysis of Ac-225, the paper doesn't provide strong storyline with enlightenment of technical realization of the technique for in vivo imaging. Experimental set-up and investigated sample are not necessarily representative and justify the conclusion; limitations are not properly addressed. Significant part of the study is focused on analysis of very low energy photons, which are not relevant for in vivo imaging.

Answer: We thank the reviewer and agree that the presented experimental setup is currently not suitable for *in vivo* imaging. The aim of the work carried out here was not to introduce a new detector technology into in vivo SPECT imaging (this still requires considerable further research and technical developments), but rather to investigate the extent to which MMC detectors are suitable for generating a significantly higher energy resolution of photons in the low-energy range than is currently available. We were able to demonstrate this impressively, which is why the results presented here lay the foundation for the future direct detection and quantification of ^{225}Ac in addition to ^{221}Fr and ^{213}Bi , giving highly accurate and reliable dosimetric data rather than using

moderately good surrogates. For this purpose, the ^{225}Ac photons at 83 and 86 keV would be of particular interest, analogous to what has recently been clinically demonstrated for $^{221}\text{Fr}/^{213}\text{Bi}$ with 78 keV and SPECT imaging (DOIs: 10.1097/RLU.0000000000002525 and 10.1007/s00259-024-06681-2). We have now elaborated on this in more detail in the manuscript:

“Before direct ^{225}Ac imaging without surrogates for *in vivo* applications can be realized further investigation of the absorption of X-rays in tissues is needed. These absorption processes depend on the patient’s mass, the location of the radionuclide in the body, and the energy of the photons. There is a need of medical physics calculations before quantification in *in vivo* experiments of the individual radionuclides are possible. To obtain a rough estimate, a simplified mouse model can be considered. Mice consist of approximately 70% water ⁴¹ and photon absorption in water varies with energy. Low energy X-ray transmission through 1 mm and 10 mm of water is plotted against photon energy in Figure S15 and Figure S16. In the low energy X-ray region around 14 keV (Figure 5), transmissions of 82% (1 mm) and 13% (10 mm) are observed. Signals above background levels remain detectable, and absorption can be corrected for. Thus, for tissue sections 2–5 μm thick, entire organs, and potentially tumor tissue in mice imaging using this region is possible. Bone and organ densities can reach up to 1.5 g/cm^3 , with corresponding absorption shown in Figure S17 and Figure S18. Beyond the centimeter scale, transmission decreases significantly for the low energy X-rays. While this can be partially offset by improved detector sensitivity, it remains a limiting factor. However, this limitation loses significance for photons with energies of 70 keV and above. This corresponds to the second energy region (Figure 4) investigated in this paper which achieve transmissions of 14% or more even through 100 mm of water. It has very recently been shown that photons in the 78 keV energy range can make important contributions to dosimetry of ^{221}Fr and ^{213}Bi in patients ^{4,42}. Future investigations will determine which energy regions in the spectrum are particularly useful for which application.”

Comment 3: The title is not precise; Term “theragnostic” includes diagnostic component and doesn’t reflect the problem statement;

Answer: The goal of this work is to detect domestic diagnostic data from the Ac-225 decay chain, which has already been successfully used in therapeutics. In the future, we aim to eliminate the need for surrogate radionuclides for diagnostics, enabling Ac-225 to be used as a true theragnostic radionuclide. As this is the beginning of our efforts, we have titled the work “Towards Cancer Theragnostic” showing that there is a way to go still before we reach that goal.

Comment 4: Abstract and later – Term “gamma-spectroscopy” might be more specific, since the paper refer to the practical application of the selected technique;

Answer: We think that with using the combination of X-ray and gamma-spectroscopy we might be even more precise since we cover a large energy region.

Comment 4: Introduction: Introduction should focus on the state-of-the-art technology, relevant for the storyline (i.e. SPECT); it is not clear, why it includes basic discussion on PET imaging or HPGe detector.

Answer: Thanks for the comment, we wanted to introduce the current gold standards for imaging radionuclide distribution (SPECT and PET) and then the current high-resolution spectroscopies like SDDs and HPGe detectors. Since we on the one hand wanted to show their limitations and on the other hand use SDDs to benchmark our measurements.

Comment 6: Line 79 – SPECT scintillators such as NaI or CsI are considered to have high sensitivity;

Answer: Thanks for the comment, indeed our phrasing was misleading, we removed it. Initially we meant that with having broad peaks smaller overlapping peaks are not as sensitively detected.

Comment 7: Line 80 – the statement is not supported by Reference #3. It might be supported by Reference #10(?);

Answer: Thanks for commenting. This statement is indeed supported by Reference #4 and not by Reference #3. The missing reference has now been added.

Comment 8: SDDs is included in the experimental setting as benchmark but is not properly introduced.

Answer: We thank the reviewer for the comment. Our answer to this question is twofold: An introduction of the SDDS is included in the introduction together with other state of the art detection methods:

“An even better energy resolution of approximately 125 eV @ 5.9 keV Mn K α line is achieved by Silicon Drift Detectors (SDDs) ¹⁴. It is based on the principle that radiation interacting with silicon generates electron–hole pairs whose electrons are guided by an internal drift field toward a small anode. The collected charge is proportional to the deposited energy and enables precise energy measurement The potential distribution is created by surface electrodes and bias voltages to small collecting anode ¹⁵. However, the energy range is typically limited to below 20–30 keV ¹⁶.”

Additionally we included details on the SDD detector in the Methods section of the Paper.

“For comparison a SDD (Amptek, FAST SDD[®], 70 mm² active area, 12.5 μ m Be-window, Bedford) was placed at a distance of a few cm from the sample and collected data for 4 h.”

Comment 9: Statement lines 102 – 105 is confusing and seems to be not relevant. It is as well not part of the subsequent discussion.

Answer: We thank the reviewer for the comment, we have removed it from the introduction and instead have written a sentence about future possibilities of MMCs in the outlook section:

“Another direction of MMC detector developments could lead to ultra-high energy resolution enabling the detection of chemical shifts. From these it will be possible to infer chemical information about the changes in the speciation of the daughter nuclides due to bond breakage or cellular interaction. This cannot be obtained with any of the currently available detector technologies.”

Comment 10: MMC introduction doesn't include information on material of the absorber and its dimension. Introduction and later discussion should properly address potential limitations. In particular please discuss sensitivity and limitations related to the size of the absorber. Potential technical limitations/ advantages such as time resolution and compatibility of MMC with imaging algorithm, how spatial resolution could be achieved.

Answer: Thank you for the comment, we agree that the missing dimensions were a mistake that needed fixing. We now included additional information in the methods part as well as in the discussion section. We agree that at this point in time one limitation is that these are quite complex measurements with the need for very low temperatures inside the cryostat. We already see that these cryostat techniques are becoming more standard due to their increasing demand for quantum computing. With this research, as for many emerging techniques there are many obstacles and potential limitations to tackle for the future, we noted in the manuscript a couple of possible ways how these can be improved in future work:

“For future work, we propose the development of a new detector for γ - and X-ray spectroscopy up to 120 keV, optimized for applications in medical physics. Such a detector will feature thicker absorbers of approximately 100 μ m or 250 μ m. To maintain the unprecedented energy resolution the lateral size of the absorbers can be reduced accordingly, because the energy resolution is proportional to the volume of the absorber. This would additionally lead to higher pixel density and thus higher spatial resolution. Furthermore, improved statistics from longer measurements can easily enhance the energy calibration and spectral analysis. Another direction of MMC detector developments could lead to ultra-high energy resolution enabling the detection of chemical shifts. From these it will be possible to infer chemical information about the changes in the speciation of the daughter nuclides due to bond breakage or cellular interaction. This cannot be obtained with any of the currently available detector technologies. Additionally, employing collimator optics in *in vivo* will facilitate high spatial resolution, enhancing the precision and accuracy of *in vivo* experiments. The well resolved spectra allow to easily detect and quantify impurities and will further improve the reliability and quality of radiation-based experiments and treatments. With ongoing development, the MMC detectors can become invaluable tools for advancing both basic research and clinical applications in nuclear medicine and radiopharmaceuticals and advancing personalized T α T.”

Comment 11: Results: Results and later discussion are dominated by observations made in very low energy spectrum. Even the results seem to be encouraging, those low energy gamma and X-

rays, will not contribute to in vivo imaging. Additional observations and other potential areas of research should be clearly differentiated from the main storyline.

Answer: We thank the reviewer for this comment, we included a detailed discussion on the limitations including calculations on a simplified mouse model and detailed the need for medical physics calculations. As detailed above the ^{225}Ac photons at 83 and 86 keV would be of particular interest for imaging, analogous to what has recently been clinically demonstrated for $^{221}\text{Fr}/^{213}\text{Bi}$ with 78 keV and SPECT imaging (DOIs: 10.1097/RLU.0000000000002525 and 10.1007/s00259-024-06681-2).

Comment 12: Discussion: Please consider comments above. Properly address all limitations. Provide outlook for further development and implementation for in vivo imaging, and other areas of research. Are follow-up experiments planned (with representative samples, collimator technique, etc)?

Answer: Even though the results shown here demonstrate for the first time that direct imaging of all nuclides in the decay chain of ^{225}Ac is possible thanks to the very high energy resolution of the photons of the individual nuclides, the reviewer is of course right in saying that, building on this very fundamental work, further research is needed before it can be used for *in vivo* imaging. We detailed further avenues of future work in the manuscript. See the paragraph copied to the answer of comment 10. Of course, we agree with the reviewer, this is only the beginning and the first step on our way to image ^{225}Ac and other nuclides many more experiments are needed and planned.